# RZcoin: Ethereum-Based Decentralized Payment with Optional Privacy Service

**DOI:** 10.3390/e22070712

**Published:** 2020-06-27

**Authors:** Hong Zhao, Xue Bai, Shihui Zheng, Licheng Wang

**Affiliations:** 1State Key Laboratory of Networking and Switching Technology, Beijing University of Posts and Telecommunications, Beijing 100876, China; marcus.zhao@tron.network (H.Z.); bupt_bx_xxaq2013@bupt.edu.cn (X.B.); shihuizh@bupt.edu.cn (S.Z.); 2Technical Department, Golden Siv Technology Limited, Level 7, K11 ATELIER Victoria Dockside, 18 Salisbury Rd, Tsim Sha Tsui, Hong Kong, China

**Keywords:** blockchain, privacy protection, zero-knowledge proof, publicly verifiable

## Abstract

As the blockchain 2.0 platform, Ethereum’s turing complete programming language and smart contract components make it play an important role in the commercialization of blockchain. With the further development of blockchain applications, the privacy and security issues of Ethereum have gradually emerged. To solve this problem, we proposed a blockchain privacy protection model called RZcash in the previous work. It implements the dynamically updateable and verifiable hiding of the asset information in Ethereum, namely the account balance and transaction amount. However, RZcash does not pay attention to the key redundancy problem that may be caused by the creation of secret accounts. In addition, the large size of proofs gives it high communication costs. In response to these problems, we further improve RZcash. For the key redundancy problem, we construct a new signature scheme based on the ciphertext equivalent test commitment. Moreover, we use the Schnorr signature and bulletproof to improve the corresponding proof scheme in RZcash, thereby reducing the size of proof. Based on these improvements, we propose a decentralized payment system, called RZcoin, based on Ethereum. Finally, we implement the algorithm model of RZcoin and evaluate its security and performance. The results show that RZcoin has higher security and Lower communication cost than RZcash.

## 1. Introduction

### 1.1. Background

With the rise of Bitcoin [1], the cryptocurrency system gradually attracted widespread attention worldwide. Since then, systems such as Ethereum [2] have begun to emerge. In June 2019, Facebook took the lead in publishing a white paper on the global cryptocurrency project called Libra [3]. According to the white paper, Libra will be the simple, borderless currency and financial infrastructure that serves billions of people around the world. The emergence of Libra has led governments to recognize the need for blockchain research and applications and start developing their own legal digital currency. However, at the recent hearing on Libra, the privacy of cryptocurrency system was questioned. The privacy disclosure has become a key issue that must be resolved in the process of legalizing the cryptocurrency.

Throughout the history of cryptocurrency, privacy has always been the design goal of these systems. From the beginning, privacy protection has been the focus of researchers. In 1982, David Chaum, known as the father of the digital currency, proposed Ecash [4]. Chaum used self-designed blind signature technology to implement an anonymous communication mechanism. Ecash became the first anonymous digital currency system to achieve untrackable, protecting users’ privacy while ensuring currency circulation. However, because it relies on the assistance of central agencies (i.e., banks) to solve problems such as double-spending, it was not widely used in the end. B-money is the first truly decentralized cryptocurrency in the world, and was proposed by Dai in 1998 [5]. It is the first to introduce the concept of PoW (Proof of Work) proposed by Adam in [6] into the issuance of digital currency. The decentralized structure helps these systems solve privacy disclosure issues caused by centralization. However, in order to achieve distributed consensus, Dai proposed that the transaction must be publicly broadcast through the whole network. It weakens the privacy of cryptocurrency system to some extent. In 2004, with the advent of RPoW (Reusable Proofs of Work) [7], the development of the cryptocurrency turned a corner. Based on B-money, Nakamoto introduces technologies such as timestamp [8,9,10,11], Merkle tree [12] and the longest chain mechanism to solve the double-spending problem. The privacy setting of cryptocurrency system that subsequently appeared (e.g., Ethereum) is basically based on Ecash and B-money. The core technology of these systems is blockchain. This concept originates from “Bitcoin: A Peer-to-Peer Cash Payment System” published by Nakamoto in the cryptography mailing list of cryptography Info Page at the end of 2008. It is essentially a distributed ledger that shares and stores data in a decentralized manner. The birth of blockchain benefits from the privacy needs of digital currencies. Its technical structure gradually formed and improved in the development process of these currency systems. However, these blockchain-based systems only cut off the connection between the identity of users and the transaction script. Sensitive information such as the asset value is still exposed to the network. In the anonymous mechanism, the transaction address is hidden by a hash function. Users can generate multiple addresses to participate in the transaction. However, due to the uniqueness of the hash value and the accuracy of the behavior analysis of users, it is possible to reconstruct the connection between the identity and the account address by analyzing a large number of transactions. Moreover, since the transaction amount is public throughout the network, malicious adversaries may concentrate on attacking transactions with high value. This will make the transaction script easily destroyed, resulting in the disclosure of important data such as business information.

The above problems not only affect the legalization of digital currency, but also hinder the application of the blockchain in the industrial Internet and other fields. China’s legal digital currency will be issued soon. The government has also begun to actively develop the blockchain technology. Therefore, it is urgent to build a blockchain system with complete privacy protection services. In response to this demand, we proposed a blockchain privacy protection solution called RZcash in 2019. It realizes the hiding of account balance and transaction amount in Ethereum by integrating the existing cryptographic technology [13]. Specifically, in RZcash, we use Pedersen commitment to achieve dynamic hiding of asset information, use the rangeproof scheme based on the two-party ring signature to provide evidence for the legality of asset commitments, and use the zero-knowledge proof based on ECDSA to ensure that secret accounts are created correctly. We proved that RZcash has good performance while ensuring security and privacy. However, due to the large size of rangeproof and zero-knowledge proof, its communication cost is higher. In addition, RZcash does not pay attention to the key redundancy problem that may be caused by the creation of secret accounts. These problems may hinder the practical application of RZcash, so it needs to be further improved.

### 1.2. Motivation and Contribution

In this regard, we proposed a series of improvements in this paper. Based on RZcash, we further improve its core algorithms and propose a decentralized payment scheme called RZcoin that provides the optional privacy protection service. When users need the service, they must create a new secret account through the creation transaction. The account can be traded with other secret accounts or with open accounts (i.e., the external account of Ethereum). The former is called the secret transaction, and the latter is called the semi-secret transaction. In general, our main contributions are as follows:We further improve the privacy protection mechanism called RZcash. For the key redundancy problem, we propose a signature scheme called CEs based on the ciphertext equivalent test scheme [14]. It achieves the creation of secret accounts without regenerating the private key. In addition, compared to ECDSA, its signature size is reduced by 80%. In addition, we use Schnorr signature to reconstruct the zero-knowledge proof of RZcash and reduce the size of proof by 1/3. We also use bulletproof to reduce the size of rangeproofs.We combine the improved privacy protection mechanism with Ethereum and propose a decentralized payment called RZcoin with optional privacy services. In RZcoin, the hidden asset can be publicly verifiable and dynamically updated. Users can neither overspend nor forge their assets. All proofs required for transactions involving the privacy service (i.e., secret and semi-secret transactions) are attached to the "extraData" module of transactions.We implement the above decentralized payment scheme based on the Ethereum test chain and evaluate its performance. At the same time, we also propose the adversarial model it faces and give the corresponding security analysis. The result shows that the scheme can provide the privacy service to users with good performance while ensuring security.

### 1.3. Related Work

By summarizing the existing research results, we observe that the privacy service structure of the blockchain system is mainly divided into two modes: off-chain and on-chain.

In the off-chain model, the private data is stored and calculated in the off-chain environment, thereby avoiding the privacy disclosure caused by the on-chain storage. For example, the Keep project initiated by Luongo et al. expands the basic application of smart contracts through the form of private data containers [15]. In addition, it establishes an access channel between Ethereum smart contracts and the private data through secure multiparty computing (SMPC). In 2017, Microsoft proposed an Ethereum-based blockchain framework called Confidential Consortium (CCF) [16]. It uses local trusted execution environments (TEEs, such as Intel SGX) to provide the secure off-chain operating and storage space for the code and data of the blockchain system. Although these two projects are still in the theoretical stage, they provide a unique idea for the privacy protection research of the blockchain.

Most of the existing research results achieved privacy protection in an on-chain mode, such as Dash [17], Monero [18], Zerocash [19] and so on. This type of system mainly implements real-time hiding of transaction information by using cryptographic technologies such as commitment. The key technology on which they are based is divided into two main categories. The one is ring signature [20]. Its typical use case is Monero. It hides the transaction amount through confidential transaction and uses a new type of ring signature called MLSAG to hide participants in the transaction [21,22,23]. The ring signature cannot only hide the source of the transaction, but also prove the correctness of hidden amount. The other is zero-knowledge proof [24]. In 2015, BenSasson et al. further developed the zk-SNARKs technology on the basis of zero-knowledge proof, which enables people to succinctly and non-interactively prove that they know certain information without revealing specific knowledge. Zcash is created based on zk-SNARKs. Subsequently, zero-knowledge proof has gradually become the mainstream technology for privacy protection. Mimblewimble [25], PLONK [26], Halo [27], Sonic [28] and other blockchain privacy protection projects were proposed. These projects are to improve the existing zk-SNARKs, in order to reduce the number of trusted settings needed for its operation, so as to improve the efficiency and universality. Although zero-knowledge proof is inefficient and difficult to implement, it is still considered to be the most effective solution to privacy issues of blockchain.

Compared to Bitcoin, the privacy requirement of Ethereum is more complex. In addition to hiding the transaction amount and participants, Ethereum also needs to hide the account balance and transaction details of the smart contract. In the past two years, Mobius [29], Zether [30], AZTEC [31], PGC [32] and other privacy protection schemes for Ethereum were proposed. However, because most of their privacy services are provided based on smart contracts, the expensive consumption of GAS prevents it from being put into practical use. In this regard, the privacy protection mechanism we proposed refers to the idea of off-chain computing. It encapsulates the encryption and verification procedures involving secret transactions in TEEs, and performs the additional calculations required for secret transactions in the off-chain environment. The privacy protection mechanism we proposed has good portability and can be combined with existing zero-knowledge virtual machines to delegate some computing work to the off-chain semi-trusted server.

### 1.4. Roadmap

The rest of this paper proceeds as follows. In Section 2, we introduce some basic knowledge used in this paper. In Section 3, we present our main originality, i.e., the ciphertext-equivalent signature (CEs) for RZcoin. In Section 4, we present the system model and the adversarial model of RZcoin. In Section 5, we introduce the interactive details of the involved roles of RZcoin. In Section 6, we give the security analysis of RZcoin according to the system model given in Section 4. In Section 7, we implement the core model of RZcoin and provide the performance analysis, especially the improvements to the corresponding proof sizes. Finally, we summarize the full text and point out the future research direction in Section 8.

## 2. Preliminary

### 2.1. Ethereum System

Ethereum is known as the blockchain 2.0 platform. It is widely used because of its turing-complete scripting language, account-based transaction model and practical smart contract components. According to the description in its white paper, the architecture of Ethereum is mainly divided into seven layers: application, contract, consensus, protocol, network, data and storage, which arranged in order from top to bottom as shown in Figure 1. Among them, the data layer mainly deals with various types of data in Ethereum. It is the gathering place of cryptographic technology and the focus of our research. Our privacy protection mechanism is mainly applied to this layer.

We can see that the main body of the data layer is the transaction. It is stored in the transaction pool and packaged into blocks by the full node after passing the legality verification. Then it is recorded in the blockchain after multi-party consensus. As shown in Table 1, the transaction format of blockchain system is basically the same, which mainly includes information such as the address of transaction participants, the transaction amount, the hash value and the signature of transaction. In addition, the transaction in Ethereum also includes the payload module. This module is mainly used to store other information required for transactions. In RZcoin, we use it to carry the extra data needed for secret transactions.

### 2.2. Number Theory Knowledge

As a basic tool in the public key cryptography, number theory played a very important role in the research and development of modern cryptography. The discrete logarithm problem (DLP) is the basic conclusion in it and is widely used in many public key cryptographic schemes. In 1985, Koblitz and Miller first proposed the creative idea of applying the elliptic curve to cryptography and constructed the discrete logarithm problem on elliptic curves (ECDLP) based on the Abelian addition group consisting of elliptic curve points [33]. The cryptographic schemes used in this paper are mainly based on ECDLP. The relevant definition is as follows.

**Definition** **1**(DLP [33])**.**
*Let g be a generator of the cyclic group G with prime order, i.e., $G=\langle g\rangle$. Randomly select h from G. If there exists a negligible function negl(λ) for all probabilistic polynomial time (PPT) adversaries A, where λ is a security parameter, such that*
(1)$$P\left[h=g^{a}\;|\;a\leftarrow A\left(g,h\right)\right]\leq negl\left(\lambda \right)$$
*then we say that it is difficult to solve the discrete logarithm problem.*

Let *E* be the elliptic curve group on the finite field $F_{q}$, where $q=p^{m}$. In this paper, we take *m* as 1. Let *G* be a generator of $E(F_{q})$. Similar to the above, the definition of ECDLP is given below.

**Definition** **2**(ECDLP [33])**.**
*Let G be the base point of group $E(F_{q})$. Randomly select point H from $E(F_{q})$. If for all PPT adversaries A, there exists a negligible function negl(λ) such that*
(2)$$P\left[H=k\cdot G\;|\;k\leftarrow A(G,H)\right]\leq negl\left(\lambda \right)$$
*then we say that it is difficult to solve the elliptic curve discrete logarithm problem.*

All privacy protection schemes used in this paper involve the point addition operation on elliptic curves. Therefore, we can provide security for our privacy protection solution based on ECDLP. For example, for the Pedersen commitment CM=s·G+r·H we used, when the adversary only knows CM, *G* and *H*, he cannot obtain the secret value *s* in polynomial time because of ECDLP.

To solve the problem of the construction of secret accounts in RZcoin, we propose the ciphertext equivalent signature scheme called CEs based on the existing ciphertext equivalent scheme. This scheme mainly relies on bilinear mapping in number theory [34]. The definition of bilinear mapping is as follows.

**Definition** **3**(Bilinear Mapping)**.**
*Given an addition cyclic group $G_{1}$ of prime order q and a multiplication cyclic group $G_{2}$, we define a mapping relationship $e:G_{1}\times G_{1}\to G_{2}$ such that it satisfies the following properties:*
Bilinear: $\forall \;P,Q\in G_{1}$, $\forall \;a,b\in Z_{q}^{*}$, $e\left(a\cdot P,b\cdot Q\right)=e{\left(P,Q\right)}^{ab}$ is established, where $Z_{q}^{*}=\left\{1,\ldots ,q-1\right\}$;Non-degenerate: $\exists \;P,Q\in G_{1}$, $e\left(P,Q\right)\neq 1\in G_{2}$ is established;Computability: $\forall \;P,Q\in G_{1}$, there is an efficient polynomial time algorithm for computing e(P,Q).
*If all the above properties are satisfied, the mapping relationship e is called a bilinear mapping.*

### 2.3. Pedersen Commitment

In this paper, we use the Pedersen commitment scheme based on elliptic curves to hide the asset information [35]. Its inherent homomorphism can prove the linear relationship between the hidden asset.

**Definition** **4**(Pedersen Commitment [35])**.**
*For the elliptic curve group $E(F_{q})$, randomly pick $G,H\overset{R}{\leftarrow }E(F_{q})$. $log_{G}^{H}$ is unknowable. The Pedersen commitment scheme contains the following three algorithms:*
$pp\leftarrow Setup(1^{\lambda })$. For security parameter $1^{\lambda }$, it outputs public parameters pp which will exist as implicit input in the following algorithms.*CM←Com(s,r). For the secret value $s\in Z_{q}^{*}$, pick the random number $r\overset{R}{\leftarrow }Z_{q}^{*}$, then output:*(3)CM=s·G+r·H{0,1}←Open(CM,s,r). If Equation (Equation 3) holds, the output of this algorithm is 1. Otherwise the output is 0.

On the basis of the above scheme, Bunz et al. proposed the Pedersen vector commitment in the bulletproof scheme published in 2018 to further strengthen the scalability of confidential transactions [36]. Bulletproof uses this scheme to achieve the hiding of the secret set s. In short, it constructs the corresponding Pedersen commitment for each element in the set of secret values, thereby achieving complete hiding and computable binding of all secret values. In the following definition, the concept of vector refers to a set of n elements.

**Definition** **5**(Pedersen Vector Commitment [36])**.**
*For the elliptic curve group $E(F_{q})$, randomly pick $\left\{G=\left(G_{1},G_{2},\ldots ,G_{n}\right),H\right\}\overset{R}{\leftarrow }E(F_{q})$, where n is the size of vector and $log_{G}^{H}$ is unknowable. A Pedersen vector commitment scheme contains the following three algorithms:*
$pp\leftarrow Setup(1^{\lambda })$. For security parameter $1^{\lambda }$, it outputs public parameters pp which will exist as implicit input in the following algorithms.*CM←Com(s,r). For the secret vector $s\in Z_{q}^{n}$, pick the random number $r\overset{R}{\leftarrow }Z_{q}^{*}$, then output:*(4)$$CM=s\cdot G+r\cdot H=r\cdot H+\sum_{i=1}^{n}s_{i}\cdot G_{i}$$*where $Z_{q}^{n}$ denotes the set of vectors containing n elements in $Z_{q}$.*{0,1}←Open(CM,s,r). If Equation (Equation 4) holds, the output is 1. Otherwise the output is 0.

It was proved that because of the hardness of ECDLP, both of these commitment schemes are unconditionally hiding and computationally binding. The definition of these properties is as follows.

**Theorem** **1**(Unconditionally Hiding [35])**.**
*If a commitment Com(s,r) will not reveal any useful information about s, it is considered to be unconditionally hiding. That is, for any PPT adversary A, his advantage satisfies:*
(5)$$Adv_{A}\left(\lambda \right)=P\left[b=b^{\prime }|\begin{array}{c} pp\leftarrow Setup(1^{\lambda });\\ \left(s_{0},s_{1}\right)\leftarrow A\left(pp\right);\\ b\overset{R}{\leftarrow }\left\{0,1\right\},r\overset{R}{\leftarrow }Z_{q}^{*},\\ CM\leftarrow Com(s_{b},r);\\ b^{\prime }\leftarrow A\left(pp,CM\right). \end{array}\right]=\frac{1}{2}$$

**Remark** **1.**
*The proof of this theorem can be referred to [35].The so-called unconditional hiding means that any two commitments are computationally indistinguishable. That is to say, even if the commitment CM is constructed by a value randomly selected from the set $\left(s_{0},s_{1}\right)$ known to the adversary, he can only guess the value contained in the CM correctly with half the probability.*


**Theorem** **2**(Computationally Binding [35])**.**
*If a commitment Com(s,r) can be opened to two different messages, ECDLP can be easily solved. That is, for any PPT adversary A, there is a negligible function negl(λ) that makes his advantage satisfy:*
(6)$$Adv_{A}\left(\lambda \right)=P\left[CM_{1}=CM_{2}|\begin{array}{c} pp\leftarrow Setup(1^{\lambda });\\ \left(s_{1},s_{2},r_{1},r_{2}\right)\leftarrow A\left(pp\right);\\ CM_{1}\leftarrow Com\left(s_{1},r_{1}\right),\\ CM_{2}\leftarrow Com\left(s_{2},r_{2}\right). \end{array}\right]\leq negl\left(\lambda \right)$$

**Remark** **2.**
*The proof of this theorem can be referred to [35]. The theorem means that the content of commitment cannot be changed after it is generated. That is to say, for the commitment $CM_{1}$ that was generated based on $s_{1}$, the adversary cannot use the different value $s_{2}$ to construct the commitment $CM_{2}$ to satisfy $CM_{1}=CM_{2}$.*


### 2.4. Zero-Knowledge Proof

As early as 1985, Goldwasser, Micali, and Rackoff came up with a groundbreaking idea called “zero-knowledge proof”. The original proof system is an interactive two-party protocol in which the prover can convince the verifier that certain statement is correct without providing the verifier with any useful information. In 1988, a non-interactive zero-knowledge proof scheme (NIZK) was proposed [37]. It extends the application of zero knowledge proof and lays a foundation for its application in blockchain.

In the existing blockchain privacy protection schemes, zero-knowledge proof is used to provide evidence for the legality of confidential transactions. Moreover, it works with the public key signature scheme to achieve accountability. Currently, zero-knowledge proof is considered to be the most effective solution to blockchain privacy issues. Therefore, for the hidden asset in RZcoin, we choose a zero-knowledge proof to provide evidence for its legitimacy.

#### 2.4.1. Ramgeproof

In RZcoin, we use bulletproof to implement rangeproofs to further reduce the size of proof [36]. Bulletproof optimizes the inner-product argument proposed by Bootle [38], reduces its communication complexity from $6log_{2}n$ to $2log_{2}n$, and constructes a non-interactive zero-knowledge proof scheme with low communication complexity based on it. By constructing the inner-product operation of the Pedersen vector commitments of two polynomials L(x) and R(x), bulletproof transforms the rangeproof of *s* equivalently into the special zero coefficient problem of inner product. In bulletproof, L denotes the Pedersen vector commitment of L(x); R denotes the Pedersen vector commitment of R(x). That is, if and only if *s* is within a certain range, the zero coefficient of $\langle L,R\rangle =\sum_{i=1}^{n}L_{i}R_{i}$ has a specific form. This construction is implemented using the homomorphic property of Pedersen vector commitment. The definition of bulletproof is as follows.

**Definition** **6**(Bulletproof [36])**.**
*For the input value a and global parameters $G,H\in E(F_{q})$, this scheme can prove that $a\in [0,2^{n}-1]$ is true for the commitment CM=a·G+r·H. The value a here corresponds to the secret value s to be hidden in our scheme. In this definition, in order to ensure comparability with the details of scheme in [36], we choose to use a to denote the secret value. It mainly contains the following three algorithms:*
$\left\{G=\left(G_{1},G_{2},\ldots ,G_{n}\right),H=\left(H_{1},H_{2},\ldots ,H_{n}\right),G,H\right\}\leftarrow Setup\left(1^{\lambda }\right)$. For security parameter $1^{\lambda }$, it outputs public parameters {G,H,G,H} which will exist as implicit input in the following algorithms. The definition of these parameters was introduced earlier.*proof←ProofGen(a,r). The prover generates the rangeproof of a according to the following steps:**(1)* Convert a to a binary string $a_{L}\in {\left\{0,1\right\}}^{n}$ and calculate $a_{R}=\left\{a_{L_{1-1}},\ldots ,a_{L_{n-1}}\right\}$;*(2)* Choose blinding vectors $s_{L},s_{R}\in Z_{q}^{n}$ and random parameters $r_{1},r_{2}\in Z_{q}^{*}$ to generate auxiliary parameters A,S.*(3)* Use Fiat-Shamir heuristic to achieve non-interaction, i.e., use the hash fuction instead of the challenge interaction of y,z.*(4)* *Construct the two linear polynomials L(x), R(x) and calculate:*(7)$$T\left(x\right)=\langle L(x),R(x)\rangle$$*(5)* Use the polynomial identity test protocol to prove that Equation (Equation 7) holds and send $proof=\{\tau_{x},\mu ,T\left(x\right),L,R\}$.{0,1}←ProofVer(proof). The verifier needs to check that if L,R∈proof are actually L(x) and R(x) and if Equation (Equation 7) holds. If the verification is passed, it means that a is indeed in the range $\left[0,2^{n}-1\right]$. Otherwise, it means that a is out of range and the commitment CM is illegal.

**Remark** **3.**
*The specific details of the above definition can be found in [36]. For example, the symbols and calculation rules involved in bulletproof are described in detail in Section 2.3. For the first four steps included in the algorithm ProofGen, the corresponding calculation details can be found in the formula (36)–(45). In addition, the details of the fifth step can be found on page 17. In addition, the related content of algorithm ProofVer can be found on page 18.*


#### 2.4.2. Proof of Account Validity

In RZcoin, if users want to use the privacy protection service, they should create a new secret account for secret transactions. Although the account information will become visible throughout the network as the secret initial transaction recorded in the blockchain, the validity of new account cannot be verified directly because its balance is hidden in the commitment. Therefore, a proof must be attached to the secret initial transaction for validity verification.

Similar to the Ethereum, the account balance is zero during the initial creation period. Only through transactions with other accounts can the balance be increased. In other words, we only need to prove that the initial balance is zero. We implement this proof operation using the zero-knowledge proof based on ECDSA in RZcash. In this paper, we use the "key-prefixed" variant of Schnorr signature to reconstruct the scheme, thereby reducing the size of proof.

**Definition** **7**(The “key-prefixed” Schnorr signature [39])**.**
*The "key-prefixed" variant of Schnorr signature is defined by the following three algorithms:*
*(x,X)←KeyGen(x). Let G be the base point of $E(F_{q})$, the public key X is generated by:*(8)X=x·G*σ←Sign(x,m). Pick the random number r, the signer calculates the random challenge c and the signature σ by the following steps. Here, Hash denotes a collision-resistant hash function. In this paper, we use SHA256 to achieve it.*(9)R=r·G,c=Hash(X,R,m),s=r+cx,(10)σ=(R,s)*{0,1}←Ver(σ,X,m). If Equation (Equation 11) holds, the signature σ is valid and the output is 1. Otherwise, the output is 0.*(11)s·G=R+c·X

Based on the above scheme, we re-explain our original zero-knowledge proof as follows. To facilitate the subsequent algorithm description, we will call this scheme Szkp. Szkp mainly contains the following three algorithms:$pp\leftarrow Setup(1^{\lambda })$. For security parameter $1^{\lambda }$, it outputs public parameters pp which will exist as implicit input in the following algorithms.z←zkpGen(x,m). For the value *x* to be proved and the auxiliary parameter *m*, this algorithm calls the schnorr signature scheme to generate the corresponding evidence *z*.
(12)(x,X)←KeyGen(x),σ←Sign(x,m)
(13)z={X,m,σ}{0,1}←zkpVer(z). For the input *z*, if 1←Ver(σ,X,m), then the verification is passed, this algorithm outputs 1. Otherwise it outputs 0.

## 3. Ciphertext Equivalent Signature Scheme

In the existing blockchain system, each account contains a pair of keys (sk,pk). It can construct any number of addresses for asset circulation based on its public key pk, but its available asset is only bound to its private key sk. Every time the user initiates a transaction, he needs to sign the transaction with his sk to ensure its non-repudiation. In RZcash, the privacy service is optional. To ensure the security of secret assets, RZcash requires that each user who uses this service must create a secret account to store encrypted assets and construct secret (or semi-secret) transactions. Correspondingly, the key pair bound to this new account should also be regenerated. It can isolate the user’s open transactions from secret transactions, thereby avoiding the privacy disclosure caused by the same public key. However, it will quickly increase the storage space required for user keys. The cost of key management will also increase. The probability of the disclosure of private keys will increase accordingly.

To solve this problem, we attempt to improve the existing ciphertext equivalence test scheme and construct a new public key signature scheme, which is called ciphertext equivalence signature scheme (CEs). The so-called ciphertext equivalence test is to determine whether two different ciphertexts contain the same plaintext information without decryption. The ciphertext equivalent test scheme we used was proposed by Zhu et al. in 2018. This work improves the PKEwET scheme proposed by Lin et al. in 2016, and uses the straight line to construct equivalent tests to obtain better performance and lower ciphertext space. The specific details of the scheme called PKEwET-L are shown in Figure 2. This figure only introduces the core algorithm of PKEwET-L. Its concept definition and implementation detailscan be found in the original paper [14]. In simple terms, this scheme uses the straight line constructed by the message to implement the ciphertext authorization and equivalent test functions. The principle is: the straight line constructed by the same message must be the same. The scheme performs four authorizations based on four different scenarios, and each authorization performs different equivalent test functions. Its application scenarios are very comprehensive. Zhu et al. conducted a detailed analysis of its correctness and security. The results show that it can achieve OW-CCA security based on CDH and IND-CCA security based on DDH respectively for two different types of attackers under the random prediction model. The specific analysis process can be found in Section 3.2.5 of [14].

Inspired by the straight line construction of this scheme, we further optimize the design of its equivalent test and propose a public key signature mechanism called CEs suitable for the blockchain system. Based on the original private key of users, CEs generates a new public key by constructing the ciphertext equivalent commitment. The public key is bound to the secret account and is used to sign and verify secret transactions. The address for receiving secret transactions is also generated based on it. CEs does not need to generate a new private key, so it can reduce the cost of key management while ensuring the non-repudiation of signatures.

**Definition** **8.**
*The ciphertext equivalence signature scheme CEs is defined as the following four algorithms:*

*$pp\leftarrow Setup(1^{\lambda })$. For security parameter $1^{\lambda }$, it outputs public parameters pp which will exist as implicit input in the following algorithms.*

*$\left(\tilde{sk},\tilde{pk}\right)\leftarrow KGen\left(sk\right)$. Let G be the base point of the elliptic curve $E(F_{q})$. $Hash_{1}$ and $Hash_{2}$ are collision-resistant hash functions. The key pair $\left(\tilde{sk},\tilde{pk}\right)$ is generated based on the original private key sk:*
(14)$$\tilde{sk}=sk$$
(15)$$\tilde{b_{1}}=Hash_{1}\left(\tilde{sk}\right),\;\tilde{b_{2}}=Hash_{2}\left(\tilde{sk}\right)$$
(16)$$\tilde{pk}=\{\tilde{b_{1}}\cdot G\;||\;\tilde{b_{2}}\cdot G\}$$

*In this paper, we implement Hash1 and Hash2 based on SHA256. Their construction methods are: $Hash_{1}=SHA256\left(0||x\right),Hash_{2}=SHA256\left(1||x\right)$. That is, we construct these two different hash functions by making simple changes to the input x.*

*$Sig\leftarrow SGen(T,\tilde{sk})$. Generate a signature for transaction T by Equation (Equation 17). Here, $\tilde{b_{1}}$ and $\tilde{b_{2}}$ must be positive numbers that are not zero, and $\tilde{b_{1}}/\tilde{b_{2}}=\tilde{b_{1}}\left({\tilde{b_{2}}}^{-1}\right)\;mod\;q.$*
(17)$$Sig=Hash_{1}\left(T\right)\left(\tilde{b_{1}}/\tilde{b_{2}}\right)\cdot G$$

*$\left\{0,1\right\}\leftarrow SVer\left(Sig,\tilde{pk}\right)$. For bilinear pairing e, verify:*
(18)$$e\left(Sig,\tilde{b_{2}}\cdot G\right)\;=e\left(\tilde{b_{1}}\cdot G,Hash_{1}\left(T\right)\cdot G\right)$$

*If Equation (Equation 18) holds, the verification of Sig is passed.*



Like the other general public key signature scheme, our scheme is unforgeable and non-repudiation. Among them, the definition of unforgeability is as follows:
**Theorem** **3**(Unforgeability)**.**
*A adversary cannot successfully forge a valid signature without knowing the signer’s private key. That is, for any PPT adversary A, there is a negligible function negl(λ) that makes his advantage satisfy:*
(19)$$Adv_{A}\left(\lambda \right)=P\left[b=1|\begin{array}{c} pp\leftarrow Setup(1^{\lambda });\\ \tilde{sk^{\prime }},\tilde{pk^{\prime }}\leftarrow A\left(pp\right),\tilde{sk^{\prime }}\neq \tilde{sk};\\ Sig^{\prime }\leftarrow SigGen\left(T,\tilde{sk^{\prime }}\right);\\ b\leftarrow SigVer(Sig^{\prime },\tilde{pk^{\prime }}). \end{array}\right]\leq negl\left(\lambda \right)$$

**Proof of Theorem** **3.**As shown in Equation (Equation 17), the generation process of the signature Sig can be divided into three parts: the hash value of the transaction $Hash_{1}\left(T\right)\in Z_{q}^{*}$, the scalar $\left(\tilde{b_{1}}/\tilde{b_{2}}\right)\in Z_{q}^{*}$ and the base point *G*. Since the transaction *T* and the point *G* are publicly known throughout the network, the scalar $\left(\tilde{b_{1}}/\tilde{b_{2}}\right)$ is the main target of attacks. In other words, there are two main methods for signature forgery attacks on CEs:(1)Attack on $\tilde{b_{1}}$ and $\tilde{b_{2}}$. For the hash function $Hash_{1}$, the adversary A needs to find a pair of collisions with the same hash value $\tilde{b_{1}}$. Similarly, A also needs to find a pair of collisions with the same hash value $\tilde{b_{2}}$ for $Hash_{2}$. That is, A needs to launch the strong collision attack on $Hash_{1}\left(T\right),Hash_{2}\left(T\right)$ at the same time and all succeed.(2)Attack on $\left(\tilde{b_{1}}/\tilde{b_{2}}\right)$. A needs to launch an attack on ECDLP to Sig.For the first attack mentioned above, we use the hash function with strong collision to achieve defense. In the implementation of CEs, we construct these two hash functions based on SHA256. By adding randomness to the input of SHA256, we implement $Hash_{1}$ and $Hash_{2}$. It was proved that the SHA256 algorithm is more secure than MD5 and SHA-1 in defending against birthday attacks and known differential attacks. In other words, SHA256 has good resistance to strong collision. Therefore, CEs can resist this attack. For the second attack, since ECDLP is a difficult problem, there is currently no effective solution to the problem. As a result, CEs can also resist such attack. □

Because CEs satisfy the unforgeability, a valid signature can only be generated by the signer himself. Therefore, CEs is non-repudiation.

## 4. Overview of RZcoin

### 4.1. System Model

RZcoin is a decentralized payment system that provides optional privacy services based on the account model blockchain. In this paper, we choose Ethereum as the underlying blockchain system to build it. Each user needs to register an Ethereum account for subsequent transactions when entering RZcoin for the first time. The user can choose to mortgage a certain amount of assets to become an accounting node (i.e., full node) to participate in mining, or can only perform daily transactions as lightweight nodes. When a transaction is initiated, the complete information of it will be broadcast across the entire network through P2P communication. Some full nodes collect these transactions and verify their validity. Valid transactions will be integrated into blocks and recorded in the blockchain through a consensus mechanism (e.g., PoW).

RZcoin mainly contains three roles, namely payer, payee and verifier. When the transaction requires privacy services, its payer and payee need to first create a new account to store and transfer secret assets. We call this account a secret account. After the creation is completed, the payer and the payee can use their secret accounts for the next secret transfer without regenerating. In RZcoin we implemented, the verifier is mainly played by the full node. When the full node receives the transaction, it will verify the evidence, and the verification. In addition, only if the verification is passed, the transaction can be uploaded to the chain. Subsequently, the payee updates its balance according to the transaction data recorded in the chain. Assuming that the node *A* is the transaction payer and the node *B* is the transaction payee, the basic workflow of RZcoin is shown in Figure 3. To highlight the subject, the figure omits the description of the process of open transactions in RZcoin.

Since the privacy service provided by RZcoin is optional, users can have both open accounts and secret accounts. The composition of transactions between different accounts is also different. Transactions between public accounts do not involve privacy services, so we do not describe them in this paper. Next, we will introduce the work that each role is responsible for in secret transactions.

-Payer: When the payer wants to initiate a secret transaction, he needs to update the current commitment of secret assets (i.e., its balance commitment) based on the amount spent in this transaction. He also generates the commitment corresponding to the transaction amount. For these commitments, he needs to provide corresponding evidence to prove its legitimacy, such as range proofs. At the same time, he will use the public key of payee to encrypt the transaction amount and corresponding random number. These ciphertexts will be sent to the payee so that he can verify the correctness of the amount. The payer packages the above data into a transaction, signs it and sends it to the network.-Verifier: For the received secret transaction, the full node will verify the validity of the transaction signature, range proofs and the updated balance commitment. If the verification is passed, the transaction will be packaged into a block and recorded in the blockchain through distributed consensus.-Payee: For the secret transaction related to him in the blockchain, the payee uses his private key to decrypt the ciphertext in it and verifies the corresponding commitment. If the verification fails, the payee will publish the evidence to the network and declare the secret transaction invalid. If the verification is successful, the payee updates his locally stored balance and random numbers.

To ensure security, the secret information owned by payer and payee must be stored in their respective TEEs, and calculations involving this information are also performed in it.

### 4.2. Adversarial Model

We define some types of attacks that RZcoin may face to lay the foundation for subsequent security analysis. RZcoin mainly faces the following three attack modes, namely balance forgery attack, signature forgery attack and over-spending attack. Next, we will introduce the principles and operations of these attacks.

#### 4.2.1. Balance Forgery Attack

When the user wants to use the privacy service, he needs to create a secret account bound to his private key and publishes the initial information of it, such as its initial balance and account address, in the form of secret initial transaction on the blockchain network. The initial balance in such transaction is expressed in the form of commitment. That is, the balance of secret account is hidden from the beginning. Since then, the balance was updated in a hidden state. No one except the account owner can know the secret balance at any time.

Since RZcoin is built on Ethereum, the setting requirements of the initial balance are the same as that of Ethereum, i.e., the initial balance must be set to zero. Since the account creation is a local operation of the user, an adversary may want to violate the above system settings to forge the account balance commitment. When the adversary sets an account balance, he may want to use any known positive integer to replace zero to generate the commitment, thereby fabricating an available asset with no legal source for himself.

#### 4.2.2. Signature Forgery Attack

For each secret transaction in the system, its initiator needs to sign it to ensure the traceability of the transaction source. This also realizes the non-repudiation of transactions. RZcoin uses CEs to generate the public key of secret accounts and sign secret transactions. In this regard, malicious adversaries may want to obtain key parameters of the signature by analyzing some public information in the Internet (e.g., a message signed with the same private key). The parameter can be used to forge the signer’s signature, thereby further forging his transaction. Using this method, the adversary may want to transfer the victim’s secret assets to his account or spend them. This is the so-called signature forgery attack.

#### 4.2.3. Over-Spending Attack

In RZcoin, The asset information involved in each secret transaction is hidden in the commitment. Except for its payer and payee, no one can know the amount of this transaction. In addition, the secret balance of the transaction participants cannot be known by other nodes in the network. Malicious adversaries may want to spend more than their account balance in a single secret transaction and forge the legal range proof to disguise the unreasonable consumption behavior. This is the so-called signature over-spending attack.

## 5. Description of RZcoin

In this section, we will systematically introduce the transaction details in RZcoin. Because the focus of our work is on privacy services, we will focus on the secret and semi-secret transactions in RZcoin to introduce the details of their construction process and entire transaction cycle. Since the use of secret services requires the construction of a new secret account, the secret initial transaction is also the focus of our description. The specific process of open transactions is omitted. We start with the definition of some notations involved in RZcoin.

### 5.1. Notations

Table 2 shows some symbols used in the description of RZcoin and their meanings. For example, the symbols *G* and *H* represent two different points of the elliptic curve group $E(F_{p})$, where *G* is the base point of the curve and *H* is any point other than the base point. It requires that the discrete logarithm $log_{G}^{H}$ of *H* relative to *G* must be unknown. $\left(\tilde{sk},\tilde{pk}\right)$ is a key pair generated by the user calling CEs when creating a secret account, where $\tilde{sk}$ represents the private key of the secret account, which is the same as the original private key of the user’s open account; $\tilde{pk}$ represents the public key of the secret account, which can be used to generate the address of secret account. ID and $\tilde{ID}$ respectively represent the addresses of open account and secret account, which are used to receive and initiate transactions. Their subscript role indicates the role played by the owner of address in this transaction, such as payer and payee. ${\tilde{B}}_{role}$ represents the secret balance corresponding to ${\tilde{ID}}_{role}$ and $B_{role}$ is the open balance. ${\tilde{A}}_{pay}$ means the transaction amount involving secret assets, $A_{pay}$ is the opposite.

For the secret balance and transaction amount, RZcoin can hide them in the commitment. ${\tilde{CM}}_{role}$ represents the commitment corresponding to ${\tilde{B}}_{role}$. ${\tilde{r}}_{role}$ represents the random number required to generate ${\tilde{CM}}_{role}$. ${\tilde{CM}}_{pay}$ represents the commitment corresponding to ${\tilde{A}}_{pay}$. ${\tilde{r}}_{pay}$ represents the random number required to generate ${\tilde{CM}}_{pay}$. For the legality of these commitments, we use bulletproof and Szkp to generate corresponding evidence. ${\tilde{R}}_{cm}$ represents the range proof corresponding to cm and ${\tilde{Z}}_{cm}$ represents the zero-knowledge proof corresponding to the cm. Some of the above variables will be stored in the transaction Tx. $\tilde{\sigma }$ and σ represent the signature of transactions initiated by the secret account and open account respectively.

### 5.2. Secret Initial Transaction

The secret initial transaction is the first transaction generated by users after calling the privacy service. Its construction depends on the algorithm called AccountGen. Its input is the user’s private key sk. Its output includes the secret initial transaction $Tx_{init}$, the secret balance $\tilde{B}$ and corresponding $\tilde{r}$ which need to be stored locally. It can be written as $\left\{Tx_{init},\tilde{B},\tilde{r}\right\}\leftarrow AccountGen\left(sk\right)$. The steps of this algorithm are as follows:(1)Call the key generation algorithm KGen in CEs to generate the key pair $\left(\tilde{sk},\tilde{pk}\right)$. At the same time, $\tilde{ID}$ will be generated based on $\tilde{pk}$. Its generation method is consistent with the same functional in Ethereum.
(20)$$\left(\tilde{sk},\tilde{pk}\right)\leftarrow KGen\left(sk\right)$$(2)Set the initial balance $\tilde{B}=0$ and randomly select $\tilde{r}$ from $Z_{q}$ for subsequent commitment construction.(3)Calculate the commitment $\tilde{CM}$ of $\tilde{B}$. To prove that $\tilde{B}$ is 0, the proof generation algorithm zkpGen in Szkp needs to be called to generate the corresponding evidence ${\tilde{Z}}_{\tilde{CM}}$. *m* here is set by the system and its value will not affect the generation of evidence.
(21)$$\tilde{CM}=\tilde{B}\cdot G+\tilde{r}\cdot H=\tilde{r}\cdot H$$
(22)$${\tilde{Z}}_{\tilde{CM}}\leftarrow zkpGen\left(\tilde{r},m\right)$$(4)Integrate the data generated in the above steps into $Tx_{init}$.
(23)$$Tx_{init}=\left\{\tilde{ID},\tilde{pk},\tilde{CM},{\tilde{Z}}_{\tilde{CM}}\right\}$$(5)Use $\tilde{sk}$ to sign this transaction, i.e., call the signature generation algorithm SGen in CEs to generate the signature ${\tilde{\sigma }}_{init}$. Attach the signature to $Tx_{init}$ and broadcast $Tx_{init}$ through the network.
(24)$${\tilde{\sigma }}_{init}\leftarrow SGen\left(Tx_{init},\tilde{sk}\right)$$
(25)$$Tx_{init}\leftarrow Tx_{init}\left|\right|{\tilde{\sigma }}_{init}$$

In the above process, $\tilde{sk}$ is bound to all assets in the secret account, namely $\tilde{B}$. The randomness of $\tilde{r}$ determines the security of secret assets. Therefore, $\tilde{sk}$, $\tilde{B}$ and $\tilde{r}$ must be stored in the TEE hardware environment of the user’s local machine to ensure security. Whenever there is a transfer of secret assets, the user needs to update the locally stored $\tilde{B}$ and $\tilde{r}$ to ensure data consistency across the network.

When $Tx_{init}$ is broadcast through the entire network, some full nodes will receive it and call the algorithm AccountVer to verify its validity. Its input is $Tx_{init}$ and its output is 1 or 0. It can be written as $\left\{0,1\right\}\leftarrow AccountVer\left(Tx_{init}\right)$. If the output is 1, it means that all verifications were passed. The full node will then package it together with other valid transactions into a new block, participate in the consensus process of the entire network and wait to be recorded in the blockchain. The steps of this algorithm are as follows:(1)Call the signature verification algorithm SVer in CEs to verify the legality of ${\tilde{\sigma }}_{init}$. If it returns 1, the verification is passed. Otherwise, the algorithm is aborted and $Tx_{init}$ is discarded.(2)Call the proof verification algorithm zkpVer in Szkp to verify ${\tilde{Z}}_{\tilde{CM}}$. If it returns 1, the verification is passed. Otherwise, the algorithm is aborted and $Tx_{init}$ is discarded.(3)If all above verifications are passed, the algorithm returns 1, otherwise 0.

The process of the secret initial transaction is shown in Figure 3. All users in the system can construct a new secret account in this way.

### 5.3. Secret Transaction

In the previous section, we briefly introduced the concept of secret transactions. The entire process of secret transactions does not involve open assets. All transaction information is hidden through privacy services. Next, we will take a specific transaction scenario as an example to introduce the specific process of secret transactions. In this scenario, node A is the payer of the transaction. It uses its secret account ${\tilde{ID}}_{A}$ to initiate this transaction. Node B is the payee of the transaction. It receives the transfer from payer through its secret account ${\tilde{ID}}_{B}$. The full node plays the role of verifier. The overall process of the transaction is shown in Figure 4. Next, we will explain in detail according to different roles.

#### 5.3.1. Payer A

The construction of secret transactions depends on the algorithm SecretPay. The payer A will call it to generate the key parameters required in the secret transaction. Its input includes ${\tilde{sk}}_{A}$, A’s current balance ${\tilde{B}}_{A}^{old}$, the corresponding ${\tilde{r}}_{A}^{old}$, the amount ${\tilde{A}}_{pay}$, ${\tilde{pk}}_{B}$, B’s current balance commitment ${\tilde{CM}}_{B}^{old}$, ${\tilde{ID}}_{A}$ and ${\tilde{ID}}_{B}$. Its output includes the secret transaction $Tx_{\tilde{pay}}$, A’s updated balance ${\tilde{B}}_{A}^{new}$ and the corresponding ${\tilde{r}}_{A}^{new}$. Among them, ${\tilde{B}}_{A}^{new}$ and ${\tilde{r}}_{A}^{new}$ are used to update the corresponding variables stored in the local TEE. $Tx_{\tilde{pay}}$ is released to the blockchain network. It can be written as $\left\{Tx_{\tilde{pay}},{\tilde{B}}_{A}^{new},{\tilde{r}}_{A}^{new}\right\}\leftarrow SecretPay\left({\tilde{ID}}_{A},{\tilde{ID}}_{B},{\tilde{sk}}_{A},{\tilde{B}}_{A}^{old},{\tilde{r}}_{A}^{old},{\tilde{A}}_{pay},{\tilde{pk}}_{B},{\tilde{CM}}_{B}^{old}\right)$. The steps of this algorithm are as follows:(1)Randomly select ${\tilde{r}}_{pay}\in Z_{q}^{*}$ to generate the commitment of ${\tilde{A}}_{pay}$, and update the locally saved variables;
(26)$${\tilde{B}}_{A}^{new}={\tilde{B}}_{A}^{old}-{\tilde{A}}_{pay}$$
(27)$${\tilde{r}}_{A}^{new}={\tilde{r}}_{A}^{old}-{\tilde{r}}_{pay}$$(2)For the updated balance ${\tilde{B}}_{A}^{new}$, ${\tilde{r}}_{A}^{new}$ is used to generate ${\tilde{CM}}_{A}^{new}$. At the same time, ${\tilde{r}}_{pay}$ is used to generate ${\tilde{CM}}_{pay}$.
(28)$${\tilde{CM}}_{A}^{new}={\tilde{B}}_{A}^{new}\cdot G+{\tilde{r}}_{A}^{new}\cdot H$$
(29)$${\tilde{CM}}_{pay}={\tilde{A}}_{pay}\cdot G+{\tilde{r}}_{pay}\cdot H$$(3)For the two commitments generated in the previous step, the proof generation algorithm ProofGen in bulletproof is used to generate the corresponding evidence.
(30)$${\tilde{R}}_{{\tilde{CM}}_{A}^{new}}\leftarrow ProofGen\left({\tilde{B}}_{A}^{new},{\tilde{r}}_{A}^{new}\right)$$
(31)$${\tilde{R}}_{{\tilde{CM}}_{pay}}\leftarrow ProofGen\left({\tilde{A}}_{pay},{\tilde{r}}_{pay}\right)$$(4)Use ${\tilde{pk}}_{B}$ to encrypt ${\tilde{A}}_{pay}$ and ${\tilde{r}}_{pay}$. This operation allows B to confirm the transaction amount and verify it with the corresponding commitment to ensure that A has not cheated. The encryption algorithm here is not limited.
(32)$$C\leftarrow Enc_{{\tilde{pk}}_{B}}\left({\tilde{A}}_{pay},{\tilde{r}}_{pay}\right)$$(5)Update the balance commitment of B and attach it to the transaction.
(33)$${\tilde{CM}}_{B}^{new}={\tilde{CM}}_{B}^{old}+{\tilde{CM}}_{pay}$$(6)Construct the secret transaction $Tx_{\tilde{pay}}$ as follows, use ${\tilde{sk}}_{A}$ to generate its signature by SGen in CEs, and finally broadcast it through the network.
$$Tx_{\tilde{pay}}=\{{\tilde{ID}}_{A},{\tilde{ID}}_{B},{\tilde{CM}}_{A}^{new},{\tilde{CM}}_{pay},$$
(34)$${\tilde{CM}}_{B}^{new},{\tilde{R}}_{{\tilde{CM}}_{A}^{new}},{\tilde{R}}_{{\tilde{CM}}_{pay}},C\}$$
(35)$$\sigma_{\tilde{pay}}\leftarrow SGen\left(Tx_{\tilde{pay}},{\tilde{sk}}_{A}\right)$$
(36)$$Tx_{\tilde{pay}}\leftarrow Tx_{\tilde{pay}}\left|\right|\sigma_{\tilde{pay}}$$

#### 5.3.2. Verifier

After receiving the transaction, the full node needs to call the algorithm SecretVer to verify the validity of $Tx_{\tilde{pay}}$. Its input includes $Tx_{\tilde{pay}}$,${\tilde{CM}}_{A}^{old}$,${\tilde{CM}}_{B}^{old}$ and ${\tilde{pk}}_{A}$. Its output is 0 or 1. It can be written as $\left\{0,1\right\}\leftarrow SecretVer\left(Tx_{\tilde{pay}},{\tilde{CM}}_{A}^{old},{\tilde{CM}}_{B}^{old},{\tilde{pk}}_{A}\right)$. Only when all the following verifications are passed, the output will be 1. The full node will package $Tx_{\tilde{pay}}$ into blocks and record it in the blockchain. The steps of this algorithm are as follows:(1)Call SVer in CEs to verify the validity of $\sigma_{\tilde{pay}}$. If it returns 1, the verification is passed and the following steps can be continued. Otherwise the transaction is discarded.(2)Call the proof verification algorithm ProofVer in bulletproof to verify ${\tilde{R}}_{{\tilde{CM}}_{A}^{new}}$ and ${\tilde{R}}_{{\tilde{CM}}_{pay}}$. If they all return 1, the verification is passed and the following steps can be continued. Otherwise the transaction is discarded.(3)Respectively verify the algebraic relationship between these balance commitments. If Equations (Equation 37) and (Equation 38) hold, the verification is passed and the following steps can be continued. Otherwise the transaction is discarded.
(37)$${\tilde{CM}}_{A}^{new}={\tilde{CM}}_{A}^{old}-{\tilde{CM}}_{pay}$$
(38)$${\tilde{CM}}_{B}^{new}={\tilde{CM}}_{B}^{old}+{\tilde{CM}}_{pay}$$(4)If all above verifications are passed, the algorithm returns 1. Otherwise it returns 0.

#### 5.3.3. Payee B

When the payee B finds $Tx_{\tilde{pay}}$ in the blockchain, he needs to call the algorithm SecretUpdate to perform the final verification process shown below. Its input is $Tx_{\tilde{pay}}$ and ${\tilde{sk}}_{B}$. Its output is the verification result. 1 means pass and 0 means failure. It can be written as $\left\{0,1\right\}\leftarrow SecretUpdate\left(Tx_{\tilde{pay}},{\tilde{sk}}_{B}\right)$. If the verification is passed, the locally stored variables will be updated. If the verification fails, B will cancel the transaction and broadcast the corresponding evidence through the entire network, namely ${\tilde{A}}_{pay}$ and ${\tilde{r}}_{pay}$. The steps of this algorithm are as follows:(1)Use ${\tilde{sk}}_{B}$ stored locally to decrypt the ciphertext *C*.
(39)$${\tilde{A}}_{pay},{\tilde{r}}_{pay}\leftarrow Dec_{{\tilde{sk}}_{B}}\left(C\right)$$(2)Use the decrypted ${\tilde{A}}_{pay}$ and ${\tilde{r}}_{pay}$ to verify the legitimacy of ${\tilde{CM}}_{pay}$. If the equation holds, the algorithm returns 1. Otherwise it returns 0.
(40)$${\tilde{CM}}_{pay}={\tilde{A}}_{pay}\cdot G+{\tilde{r}}_{pay}\cdot H$$

### 5.4. Semi-Secret Transaction

The semi-secret transactions refers to transactions involving one of the payer and payee using public accounts. It realizes the conversion between secret assets and open assets. When the payer and payee are the same user, this conversion realizes the "cash-out" and "recharge" operations of the user’s secret assets. Next, we will use two specific scenarios as examples to show the interactive process of semi-secret transactions, which are called Scene 1 and Scene 2 respectively. Scene 1 is used to describe it initiated by the open account and Scene 2 is used to describe it initiated by the secret account. There are three nodes involved in these two scenarios: A, B and C. A plays the role of payer in Scene 1. It uses the open account to initiate the semi-secret transaction $Tx_{pay1}$. B plays the role of payee in Scene 1. Subsequently, B plays the role of payer in Scene 2. It uses the secret account to initiate the semi-secret transaction $Tx_{pay2}$. C plays the role of payee in Scene 2. In these two scenarios, the full node is responsible for verifying transactions. The interactive process of these scenarios is shown in Figure 5. Next, we will first introduce Scene 1 according to different roles.

#### 5.4.1. Scene 1

**Payer A**: 

The construction of $Tx_{pay1}$ depends on the algorithm open2secretPay. The payer A will call it to generate the key parameters required in this transaction. It can be written as $Tx_{pay1}\leftarrow open2secretPay(A_{pay1},sk_{A},B_{A}^{old},ID_{A},{\tilde{pk}}_{B},$
${\tilde{CM}}_{B}^{old},{\tilde{ID}}_{B})$. The steps of this algorithm are as follows:(1)Randomly select ${\tilde{r}}_{pay1}\in Z_{q}^{*}$ for the update operation of the random number saved by B. Since the transaction amount is a public value, only the commitment corresponding to ${\tilde{r}}_{pay1}$ and its zero-knowledge proof need be generated.
(41)$${\tilde{CM}}_{{\tilde{r}}_{pay1}}={\tilde{r}}_{pay1}\cdot H$$
(42)$${\tilde{Z}}_{{\tilde{r}}_{pay1}}\leftarrow zkpGen\left({\tilde{r}}_{pay1},m\right)$$(2)Use ${\tilde{pk}}_{B}$ to encrypt ${\tilde{r}}_{pay1}$ to get the ciphertext $C_{1}$. The ciphertext is sent to the B together with $Tx_{pay1}$, assisting B to update the local random number.
(43)$$C_{1}\leftarrow Enc_{{\tilde{pk}}_{B}}\left({\tilde{r}}_{pay1}\right)$$(3)Update ${\tilde{CM}}_{B}^{old}$ based on $A_{pay1}$.
(44)$${\tilde{CM}}_{B}^{new}={\tilde{CM}}_{B}^{old}+A_{pay1}\cdot G+{\tilde{r}}_{pay1}\cdot H$$(4)A update his open balance $B_{A}^{old}$.
(45)$$B_{A}^{new}=B_{A}^{old}-A_{pay1}$$(5)Construct $Tx_{pay1}$ as follows, use $sk_{A}$ to generate the corresponding transaction signature, and finally broadcast $Tx_{pay1}$ through the entire network. SigGen is the original signature algorithm of the blockchain system (generally ECDSA).
$$Tx_{pay1}=\{ID_{A},ID_{B},B_{A}^{new},A_{pay1},{\tilde{CM}}_{B}^{new},$$
(46)$${\tilde{CM}}_{{\tilde{r}}_{pay1}},{\tilde{Z}}_{{\tilde{r}}_{pay1}},C_{1}\}$$
(47)$$\sigma_{pay1}=SigGen\left(Tx_{pay1},sk_{A}\right)$$
(48)$$Tx_{pay1}\leftarrow Tx_{pay1}\left|\right|\sigma_{pay1}$$

**Verifier**:

After receiving the transaction $Tx_{pay1}$, the full node needs to call the algorithm open2secretVer to verify the validity of the transaction. It can be written as $\left\{0,1\right\}\leftarrow open2secretVer\left(Tx_{pay1},pk_{A},B_{A}^{old},{\tilde{CM}}_{B}^{old}\right)$. Only when its output is 1, can $Tx_{pay1}$ be finally recorded in the blockchain. The steps of this algorithm are as follows:(1)Call SigVer to verify the legality of $\sigma_{pay1}$, if it returns 1, the verification is passed. Otherwise the transaction is discarded.(2)Call zkpVer to verify the legality of ${\tilde{Z}}_{{\tilde{r}}_{pay1}}$. If it returns 1, the verification is passed. Otherwise the transaction is discarded;(3)Verify the correctness of the balance updates of A and B. If Equations (Equation 49) and (Equation 50) hold, the verification is passed. Otherwise the transaction is discarded.
(49)$$B_{A}^{new}=B_{A}^{old}-A_{pay1}$$
(50)$${\tilde{CM}}_{B}^{new}={\tilde{CM}}_{B}^{old}+A_{pay1}\cdot G+{\tilde{CM}}_{{\tilde{r}}_{pay1}}$$

**Payee B**: 

When B finds that $Tx_{pay1}$ is recorded in the blockchain, it calls the algorithm open2secretUpdate to update the locally stored variables. It can be written as $\left\{{\tilde{B}}_{B}^{new},{\tilde{r}}_{B}^{new}\right\}\leftarrow open2secretUpdate\left(Tx_{pay1},{\tilde{sk}}_{B},{\tilde{B}}_{B}^{old},{\tilde{r}}_{B}^{old}\right)$. The steps of this algorithm are as follows:(1)Use the locally saved ${\tilde{sk}}_{B}$ to decrypt the ciphertext $C_{1}$ to obtain the random number corresponding to the transaction.
(51)$${\tilde{r}}_{pay1}\leftarrow Dec_{{\tilde{sk}}_{B}}\left(C_{1}\right)$$(2)Use the decrypted ${\tilde{r}}_{pay1}$ to update the locally stored variables.
(52)$${\tilde{r}}_{B}^{new}={\tilde{r}}_{B}^{old}+{\tilde{r}}_{pay1}$$
(53)$${\tilde{B}}_{B}^{new}={\tilde{B}}_{B}^{old}+A_{pay1}$$

#### 5.4.2. Scene 2

In Scene 1, we describe the conversion from open assets to secret assets. Next, we will introduce Scene 2 according to different roles.

**Payer B**: 

The construction of $Tx_{pay2}$ depends on the algorithm secret2openPay. The payer B will call it to generate the key parameters required in this transaction. It can be written as $\left\{Tx_{pay2},{\tilde{B}}_{B}^{new},{\tilde{r}}_{B}^{new}\right\}\leftarrow secret2openPay({\tilde{ID}}_{B},{\tilde{ID}}_{C},A_{pay2},{\tilde{sk}}_{B},{\tilde{B}}_{B}^{old},{\tilde{r}}_{B}^{old},pk_{recC},$
$B_{C}^{old})$. The steps of this algorithm are as follows:(1)Randomly select ${\tilde{r}}_{pay2}\in Z_{q}^{*}$ to update the balance and random number of B’s secret account stored locally.
(54)$${\tilde{B}}_{B}^{new}={\tilde{B}}_{B}^{old}-A_{pay2}$$
(55)$${\tilde{r}}_{B}^{new}={\tilde{r}}_{B}^{old}-{\tilde{r}}_{pay2}$$(2)Generate the commitment of ${\tilde{r}}_{pay2}$ and its corresponding zero-knowledge proof.
(56)$${\tilde{CM}}_{{\tilde{r}}_{pay2}}={\tilde{r}}_{pay2}\cdot H$$
(57)$${\tilde{Z}}_{{\tilde{r}}_{pay2}}\leftarrow zkpGen\left({\tilde{r}}_{pay2},m\right)$$(3)Generate the updated balance and commitment.
(58)$${\tilde{CM}}_{B}^{new}={\tilde{B}}_{B}^{new}\cdot G+{\tilde{B}}_{B}^{new}\cdot H$$
(59)$$B_{C}^{new}=B_{C}^{old}+A_{pay2}$$(4)Call bulletproof to generate the corresponding evidence of the updated commitment.
(60)$${\tilde{R}}_{{\tilde{CM}}_{B}^{new}}\leftarrow ProofGen\left({\tilde{B}}_{B}^{new},{\tilde{r}}_{B}^{new}\right)$$(5)Construct $Tx_{pay2}$ as follows, use ${\tilde{sk}}_{B}$ to generate the transaction signature, and finally broadcast $Tx_{pay2}$ through the entire network.
$$Tx_{pay2}=\{{\tilde{ID}}_{B},ID_{C},{\tilde{CM}}_{B}^{new},A_{pay2},B_{C}^{new},$$
(61)$${\tilde{R}}_{{\tilde{CM}}_{B}^{new}},{\tilde{CM}}_{{\tilde{r}}_{pay2}},{\tilde{Z}}_{{\tilde{r}}_{pay2}}\}$$
(62)$$\sigma_{pay2}\leftarrow SGen\left(Tx_{pay2},{\tilde{sk}}_{B}\right)$$
(63)$$Tx_{pay2}\leftarrow Tx_{pay2}\left|\right|\sigma_{pay2}$$

**Verifier**:

After receiving $Tx_{pay2}$, the full node needs to call the algorithm secret2openVer to verify the validity of the transaction. It can be written as $\left\{0,1\right\}\leftarrow secret2openVer\left(Tx_{pay2},{\tilde{pk}}_{B},{\tilde{CM}}_{B}^{old},{\tilde{B}}_{C}^{old}\right)$. Only when the output is 1, can $Tx_{pay2}$ be finally recorded in the blockchain.

(1)Call SVer to verify the validity of $\sigma_{pay2}$. If it returns 1, the verification is passed. Otherwise the transaction is discarded.(2)Call zkpVer to verify the validity of ${\tilde{Z}}_{{\tilde{r}}_{pay2}}$. If it returns 1, the verification is passed. Otherwise the transaction is discarded.(3)Call ProofVer to verify the validity of ${\tilde{R}}_{{\tilde{CM}}_{B}^{new}}$. If it returns 1, the verification is passed. Otherwise the transaction is discarded.(4)Verify the correctness of the updated balance of B and C. If Equations (Equation 64) and (Equation 65) hold, the verification is passed. Otherwise the transaction is discarded.
(64)$$B_{C}^{new}=B_{C}^{old}+A_{pay2}$$
(65)$${\tilde{CM}}_{B}^{new}={\tilde{CM}}_{B}^{old}-A_{pay2}\cdot G-{\tilde{CM}}_{{\tilde{r}}_{pay2}}$$

**Payee C**: 

When C finds that $Tx_{pay2}$ was recorded in the blockchain, it only needs to update the locally stored balance of its open account. This operation has nothing to do with the secret service and will not be described here.

## 6. Security Analysis

In the previous article, we defined three attack models that RZcoin may face. In this section, we will perform the heuristic analysis of security of RZcoin based on these attack models.

### 6.1. Balance Forgery Attack

This attack mainly refers to an attack against the initial balance. As long as the initial balance is legal, RZcoin will ensure the legality of balances updated subsequently. RZcoin chooses to use the non-interactive zero-knowledge proof scheme called Szkp to resist this attack. This scheme can prove that the user honestly calculated the initial balance commitment of the secret account without exposing the secret information $\tilde{CM}=\tilde{r}\cdot H$. This solution is based on the schnorr signature, so its security is based on the security of schnorr signature. The schnorr signature scheme we used is the so-called "key-prefxed" variant. This variant is considered to have better multi-user security than the classic variant and is therefore widely used [40].We chose this variant to make our solution more portable. The schnorr signature has higher security than ECDSA, so the security of our scheme is also improved.

Because of the schnorr signature, the proof generated by Szkp also satisfies unforgeability. This property is based on the ECDLP. Based on it, a malicious adversary cannot know the discrete logarithm $log_{H}^{{\tilde{CM}}^{\prime }}$ of the forged commitment ${\tilde{CM}}^{\prime }$ relative to *H*, where ${\tilde{CM}}^{\prime }={\tilde{B}}^{\prime }\cdot G+{\tilde{r}}^{\prime }\cdot H$, without knowing the discrete logarithm $log_{G}^{H}$. That is, he cannot transform ${\tilde{CM}}^{\prime }$ into ${\tilde{CM}}^{\prime }=\left({\tilde{B}}^{\prime }log_{G}^{H}+{\tilde{r}}^{\prime }\right)\cdot H$. Therefore, the adversary cannot construct a legal zero-knowledge proof. This shows that **RZcoin can resist the balance forgery attack**.

### 6.2. Signature Forgery Attack

The signature mechanism of RZcoin is implemented through CEs. In Section 3, We analyzed its unforgeability. We divided the attacks against this scheme into two categories, and gave heuristic security analysis separately. We use CEs to separate the signature mechanism of secret transactions from that of open transactions. In this case, even if the signature of open transactions is attacked, as long as the private key is not leaked, the security of the signature of secret transactions will not be affected. Therefore, **RZcoin can resist the signature forgery attacks**.

### 6.3. Over-Spending Attack

In RZcoin, the asset information involved in each secret transaction is hidden in the commitment. Except for payer and payee, no node can know the transaction amount. To ensure the validity of secret transactions, we require the payer to provide the legality proof for its hidden asset information. Therefore, a malicious adversary who wants to launch an over-spending attack must be able to forge a legal proof to disguise his unreasonable consumption behavior. The security proof of bulletproof used in this article was given in [36]. **Therefore, RZcoin can resist the over-spending attacks**.

## 7. Simulation and Performance Evaluation

We use python to simulate RZcoin and test its performance. We will first evaluate the underlying improvements, such as CEs and Szkp. Then we will evaluate the performance of each algorithm in RZcoin from the running time and memory consumption. We will start with the configuration of simulation environment.

### 7.1. Configuration of Simulation Environment

To facilitate testing, we chose to implement RZcoin’s core algorithm on the laptop with Ubuntu 18.04.2 LTS operating system. The hardware and software configuration of the entire test environment is shown in Table 3. In addition, the elliptic curve point multiplication and point addition operations in RZcoin are implemented through the python version of ECDSA algorithm library. The type of elliptic curve we selected is secp256k1, and the parameter configuration of this curve is shown in Table 4. CEs is based on the pypbc library to achieve the pairing operation required for the generation and verification of signatures. For all basic operations in RZcoin, we choose to use the GNU multiple precision operation library (GMP) to ensure the accuracy of large number operations in this mechanism [41]. In addition, all large number operations are implemented in the prime field. For this field, we require the size of the prime to be 256 bits and the same order as the curve scp256k1. To ensure the safety of RZcoin, all random numbers are generated by calling the random algorithm library that comes with python and the precision is set to 256 bits.

### 7.2. Evaluation of Improvements

In RZcoin, we mainly give three improvements: (1) We propose CEs to solve the key redundancy problem caused by secret accounts; (2) We use Schnorr to reconstruct the zero-knowledge proof scheme to reduce the size of proof; (3) We used bulletproof to further reduce the size of the transaction. In the previous section, we performed a heuristic analysis of their security. It shows that these improvements improved the security of RZcoin. In this section, we will further evaluate their performance. To facilitate evaluation, the accuracy of the running time of all algorithms is set to six decimal places.

#### 7.2.1. Evaluation of CEs

The signature scheme mainly contains three algorithms, namely KeyGen, SigGen, and SigVer. We will evaluate the running time of these three algorithms and the size of the generated signature. The test results are shown in Table 5, where all the data are the average of the results obtained after 20 tests. Since the signature of CEs is a single value, its size is much smaller than that of ECDSA. Compared to ECDSA, its signature size is reduced by 80%. However, in terms of running time, CEs are not as good as ECDSA. However, the design goal of CEs is to solve the security risks caused by key redundancy. In RZcoin, two different types of accounts use different signature keys and mechanisms. Even if the signature mechanism of the one is attacked, the security of the other account can be guaranteed as long as the private key is not stolen. This setting will improve the security of the account to a certain extent. Since the running time is at the millisecond level, the difference in performance between the two algorithms is not very large. In addition, KeyGen only runs once when the account is created, so it has little impact on the system’s daily performance.

#### 7.2.2. Evaluation of Szkp

The purpose of using schnorr signatures to achieve improvements is mainly as follows: (1) Schnorr has higher security than ECDSA. Using it to achieve optimization can further enhance the security of the solution. (2) The size of the signature generated by schnorr is smaller than that of ECDSA. Therefore, this improvement can further reduce the size of transaction. (3) The schnorr signature is linear, so the zero-knowledge proof based on it also has this property. It provides the possibility for subsequent scalability improvements. We will test the proof generation and verification algorithm of Szkp and compare the results with the original scheme ZKP in RZcash. The test results are shown in Table 6, where all the data are the average of the results obtained after 20 tests. It can be seen from Table 6 that the size of proof generated by Szkp is about 2/3 of that of ZKP. In addition, the difference between the two schemes in verification time is not very large.

#### 7.2.3. Evaluation of Bulletproof

We use bulletproof to reduce the size of range proofs. Therefore, we mainly focus on the comparison between it and the range proof scheme RP of RZcash in the size of the generated proof. We use *N* to denote the maximum bit length of the object (ie number) allowed by the range proof. In its performance test, we will gradually adjust N from 2 to 64 to evaluate the influence of *N* on the size of returned result. The test results are shown in Table 7, where all the data are the average of the results obtained after 20 tests. As shown in Table 7, the proof size of bulletproof is smaller than RP, and as N increases, the difference between them becomes larger and larger. Therefore, bulletproof effectively reduces the size of proof.

### 7.3. Performance Evaluation

The above evaluation results of these improvements show that although they increase the running time to a certain extent, the size of generated proof (or signature) is much smaller than that before optimization. In addition, the security of RZcoin has also been improved. Overall, these are some good improvements. However, by simplifying RZcoin’s model algorithms, we achieved good performance in the operation of the overall system. The algorithm description for RZcoin is shown in Table 8. In Scane2, payee does not need to perform any operations related to privacy services. He only needs to update the public balance saved locally. Therefore, RZcoin does not contain related algorithms. In the model, we use *N* to denote the maximum bit length of the object (ie number) allowed by the range proof. In the performance test of RZcoin’s algorithm, we will gradually adjust N from 2 to 64 to evaluate the influence of *N* on the algorithm running time and the size of returned result. These data are the average of the results obtained after 20 tests. To facilitate evaluation, the accuracy of the calculation time of all algorithms is set to six decimal places.

Figure 6 shows the test results of the running time of AccountGen and AccountVer involved in the secret initial transaction. At the same time, for the results returned by AccountGen (specifically transactions), Table 9 also shows its memory usage. It can be seen that the running time of these two algorithms is not affected by the change of *N*. The size of secret initial transactions will also not change as *N* grows.

Figure 7 shows the test results of the running time of SecretPay, SecretVer and SecretUpdate involved in the secret transaction. At the same time, Table 10 also shows the memory usage of $Tx_{\tilde{pay}}$. It can be seen that the running time of SecretPay and SecretVer increases with the increase of *N*. However, the running time of SecretUpdate is not affected by the change of *N*. The size of secret transactions will gradually increase as *N* grows.

Figure 8 shows the test results of the running time of open2secretPay, open2secretVer, open2secretUpdate, secret2openPay and secret2openVer involved in the semi-secret transaction. At the same time, Table 11 also shows the memory usage of $Tx_{pay1}$ and $Tx_{pay2}$. The running time of secret2openPay and secret2openVer increases with the increase of *N*. However, the running time of open2secretPay, open2secretVer and open2secretUpdate are not affected by the change of *N*. In addition, $Tx_{pay2}$ will gradually increase as *N* grows, but $Tx_{pay1}$ will not change with *N*.

By summarizing and comparing the above test results, we observe that as long as the payer uses his secret account to initiate transactions, the size of transactions will increase with the growth of *N*, regardless of the account used by the payee. The running time of the generation algorithm of secret transactions that the payer needs to call and the verification algorithm of secret transactions that the full node needs to call will also show a positive growth trend with the change of *N*. The running time of all algorithms is in the millisecond level. This shows that RZcoin has good performance.

## 8. Conclusions

Based on our previous work, called RZcash, we will further improve its core algorithm and propose a decentralized payment called RZcoin that provides the optional privacy protection service. For the key redundancy caused by the secret account, we propose a signature scheme called CEs based on the ciphertext equivalent test scheme. We also use Schnorr signature and bulletproof to reduce the size of proof required for services. We implement the above decentralized payment scheme based on the Ethereum test chain and evaluate its performance. At the same time, we also propose the adversarial model it faces and give the corresponding security analysis. The result shows that the scheme can provide the privacy service to users with good performance and lower communication cost while ensuring security. Next, we will further explore how the privacy protection scheme for smart contracts can be combined with RZcoin.

## Figures and Tables

**Figure 1 entropy-22-00712-f001:**
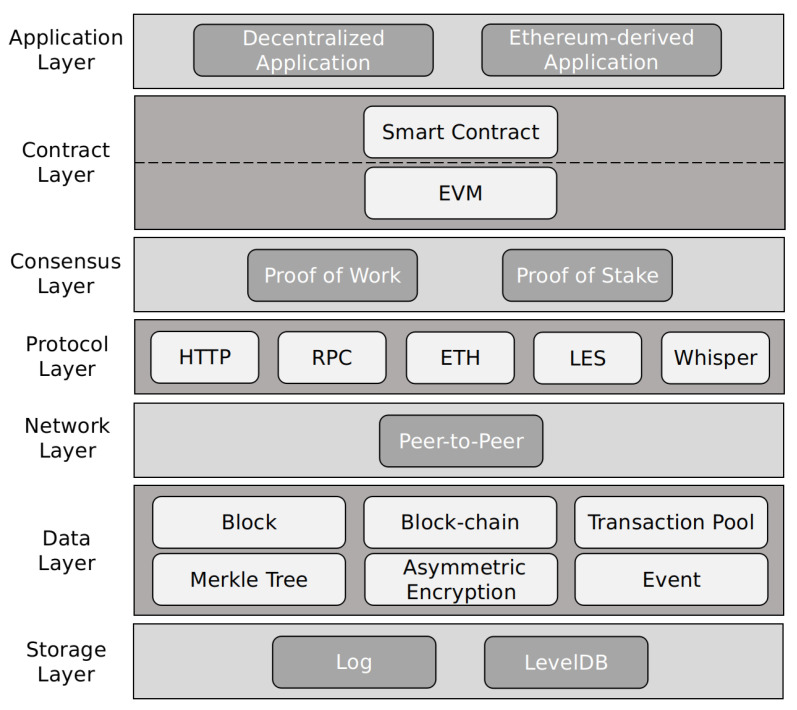
Ethereum Architecture.

**Figure 2 entropy-22-00712-f002:**
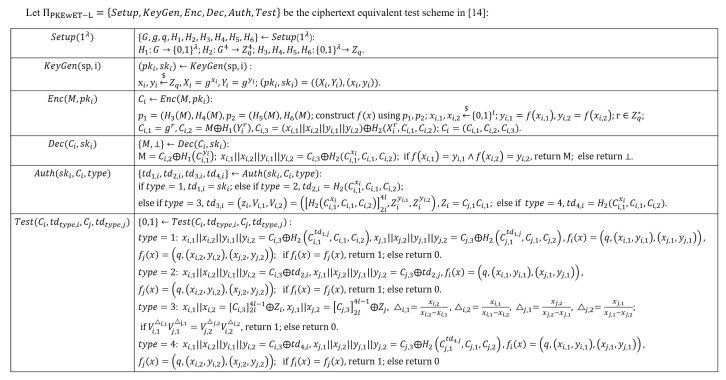
Algorithmic Composition of PKEwET-L in [14].

**Figure 3 entropy-22-00712-f003:**
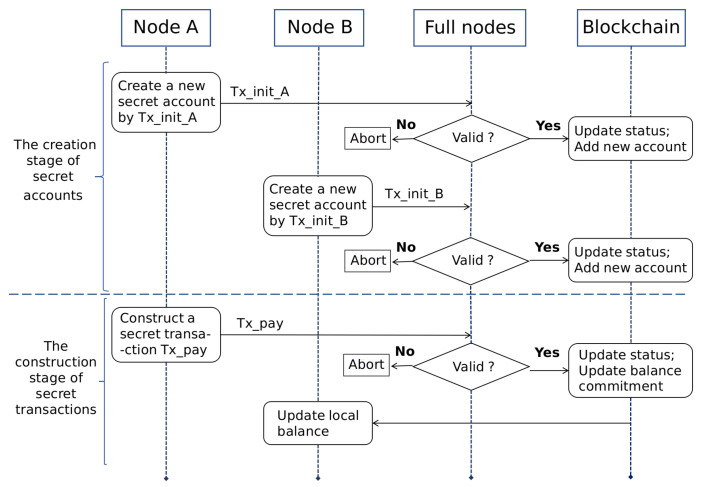
The Workflow of RZcoin.

**Figure 4 entropy-22-00712-f004:**
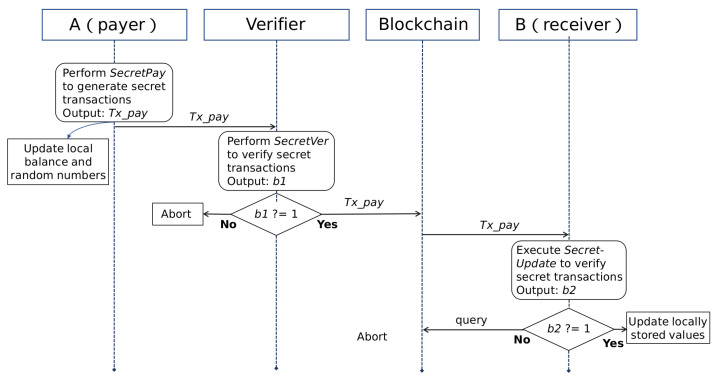
The Interactive Process of Secret Transactions.

**Figure 5 entropy-22-00712-f005:**
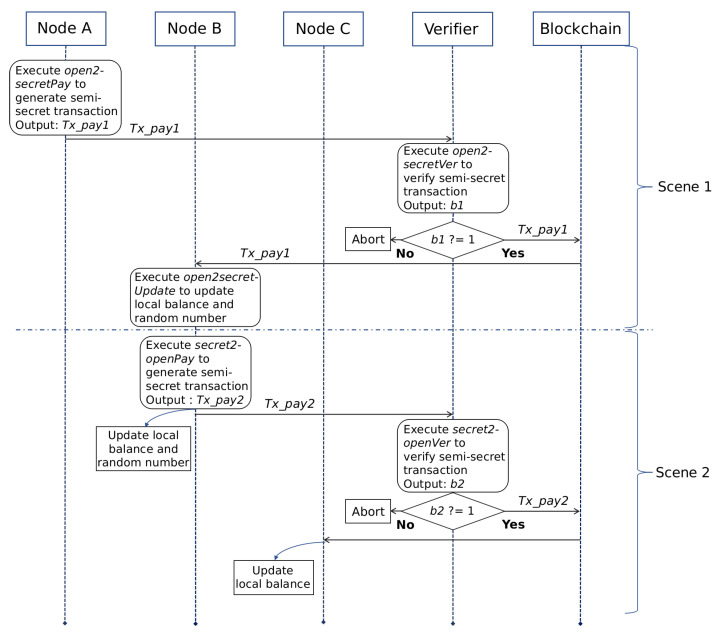
The Interactive Process of Semi-Secret Transactions.

**Figure 6 entropy-22-00712-f006:**
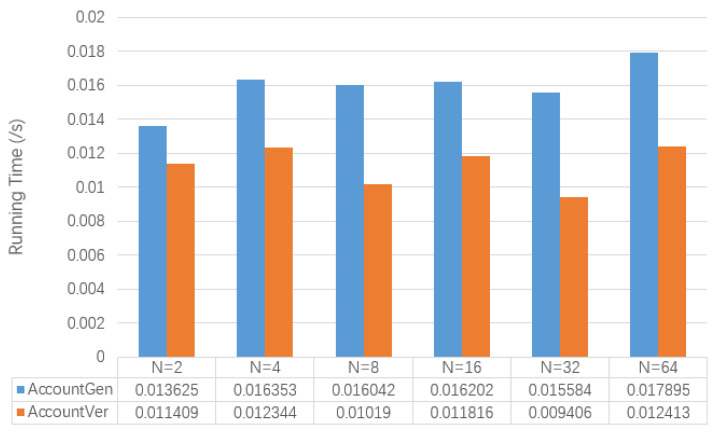
The Comparison of Algorithm Performance of Secret Initial Transactions.

**Figure 7 entropy-22-00712-f007:**
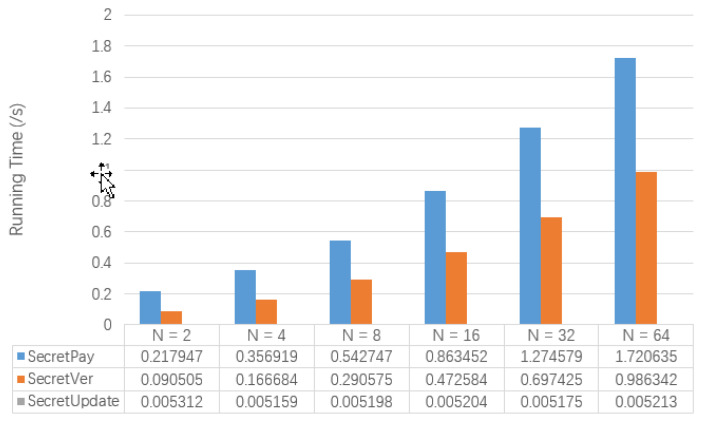
The Comparison of Algorithm Performance of Secret Transactions.

**Figure 8 entropy-22-00712-f008:**
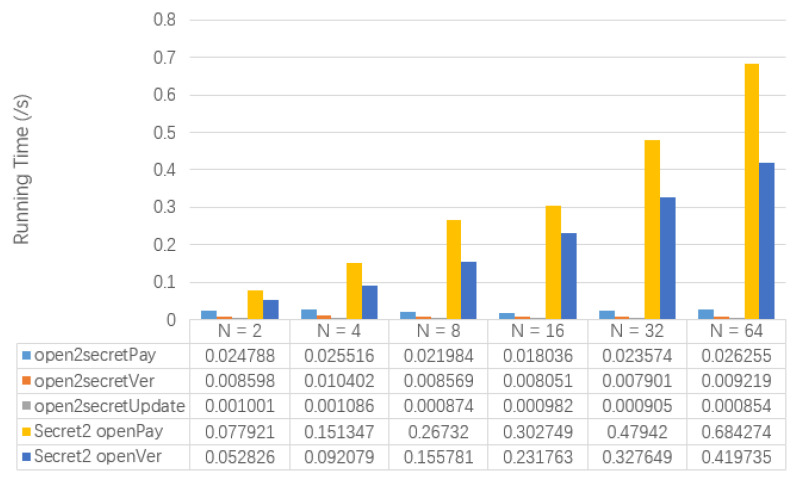
The Comparison of Algorithm Performance of Semi-secret Transactions.

**Table 1 entropy-22-00712-t001:** Transaction Format of the Blockchain System.

Transaction	
Hash	Hash value of the transaction
AccountNonce	Total number of transactions initiated by sender
GasInfo	Information about GAS of the transaction
Sender	Address of the transaction sender
Recipient	Address of the transaction recipient
Amount	Transaction amount
Timestamp	Creation time of the transaction
Signature	Signature data for the transaction
Payload	Other data of the transaction

**Table 2 entropy-22-00712-t002:** System Symbols and Their Meaning.

Notation	Description
$F_{p}$	Finite field with order *p*
G,H	Two points of $E(F_{p})$, where *G* is the base point
(sk,pk)	Key pair of open accounts
$\left(\tilde{sk},\tilde{pk}\right)$	Key pair of secret accounts
$ID_{role}$	Address of open account of role
${\tilde{ID}}_{role}$	Address of secret account of role
$B_{role}$	Open balance of role
${\tilde{B}}_{role}$	Secret balance of role
${\tilde{r}}_{role}$	Random number required for ${\tilde{B}}_{role}$
${\tilde{CM}}_{role}$	Commitment of ${\tilde{B}}_{role}$
$A_{pay}$	Transaction amount involving open assets
${\tilde{A}}_{pay}$	Transaction amount involving secret assets
${\tilde{r}}_{pay}$	Random number required for ${\tilde{A}}_{pay}$
${\tilde{CM}}_{pay}$	Commitment of ${\tilde{A}}_{pay}$
${\tilde{R}}_{cm}$	Range proof of cm
${\tilde{Z}}_{cm}$	Zero-knowledge proof of cm
$\tilde{\sigma }$	Signature generated by secret accounts
σ	Signature generated by open accounts

**Table 3 entropy-22-00712-t003:** The Hardware and Software Configuration of RZcoin’s Test Environment [41].

Name	Configuration
CPU	2.60GHz Intel(R) Core (TM) i7-6500U CPU
OS	Ubuntu 18.04.2 LTS
RAM	8 GB
python	3.7.4
pypbc	2.1.0

**Table 4 entropy-22-00712-t004:** The Parameter Configuration of Curve secp256k1.

Parameter	Recommended Value
*p*	FFFFFFFF FFFFFFFF FFFFFFFF FFFFFFFF FFFFFFFF FFFFFFFF FFFFFFFE FFFFFC2F
*a*	00000000 00000000 00000000 00000000 00000000 00000000 00000000 00000000
*b*	00000000 00000000 00000000 00000000 00000000 00000000 00000000 00000007
*G*	02 79BE667E F9DCBBAC 55A06295 CE870B07 029BFCDB 2DCE28D9 59F2815B 16F81798
*N*	FFFFFFFF FFFFFFFF FFFFFFFF FFFFFFFE BAAEDCE6 AF48A03B BFD25E8C D0364141
*h*	01

**Table 5 entropy-22-00712-t005:** Comparison of Performance between CEs and ECDSA.

	CEs	ECDSA
Time of KeyGen (/s)	0.010977	0.000087
Time of SigGen (/s)	0.001926	0.000452
Time of SigVer (/s)	0.004263	0.001930
Size of Signature (/byte)	56	296

**Table 6 entropy-22-00712-t006:** Comparison of Performance between Szkp and ZKP.

	Szkp(RZcoin)	ZKP(RZcash)
Time of proofGen (/s)	0.008463	0.000532
Time of proofVer (/s)	0.005645	0.002015
Size of zkproof (/byte)	200	304

**Table 7 entropy-22-00712-t007:** Comparison of Performance between Bulletproof and RP.

N	2	4	8	16	32	64
**Bulletproof (/byte)**	1694	2046	2279	2521	2760	3012
**RP (/byte)**	1604	3200	6376	12764	25598	51134

**Table 8 entropy-22-00712-t008:** The Algorithm Description for RZcoin.

Name	Function	Caller
AccountGen	used to build the secret account	users
AccountVer	used to verify $Tx_{init}$	full nodes
SecretPay	used to build secret transactions	payer
SecretVer	used to verify secret transactions	full nodes
SecretUpdate	used for local updates	payee
open2secretPay	used to construct semi-secret transactions in Scane1	payer
open2secretVer	used to verify semi-secret transactions in Scane1	full nodes
open2secretUpdate	used for local updates	payee
secret2openPay	used to construct semi-secret transactions in Scane2	payer
secret2openVer	used to verify semi-secret transactions in Scane2	full nodes

**Table 9 entropy-22-00712-t009:** The Size of Secret Initial Transactions.

**N**	2	4	8	16	32	64
$Tx_{init}$(/byte)	450	449	452	450	454	452

**Table 10 entropy-22-00712-t010:** The Size of Secret Transactions.

**N**	2	4	8	16	32	64
$Tx_{\tilde{pay}}$(/byte)	4036	4740	5206	5690	6168	6672

**Table 11 entropy-22-00712-t011:** The Size of Semi-secret Transactions.

**N**	2	4	8	16	32	64
$Tx_{pay1}$(/byte)	890	906	889	894	890	892
$Tx_{pay2}$(/byte)	2344	2696	2929	3171	3410	3662

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
