# Peer review of "RZcoin: Ethereum-Based Decentralized Payment with Optional Privacy Service"

_entropy, 2020, doi:10.3390/e22070712_

Round 1

Reviewer 1 Report

  • The author's work heavily depends on Schnorr algorithm. I would like authors to clearly state what is their originality other than simple application of Schnorr algorithm instead of ECDSA.
  • They did not provide proofs for Theorems 1 and 2.
  • Many definitions and many equations are provided, but they should clarify what part of them belongs to them.
  • It would be more helpful for readers to follow the defintions and theorems if they provide simple examples.

Reviewer 2 Report

Some key notions are not defined correctly and not clear, and therefore there is a problem to estimate the originality and novelty of the paper.

The paper is not well structured and a lot of notions are not introduced:

In line 191: Zq* should be exactly defined.

In line 221: what is commitment vector?

In line223: what is <L(x),R(x)>?

In line 225: are there any relation between Zq and Fp?

In line 235: equation (??).

In line 284: Hash1 and Hash2 must be exactly defined. Wihout that b1~/b2~ seems as nonsense but it is partially explained in line 289.

In line 298: is mentioned that Hash1 and Hash2 are construct on SHA-256 so this should be described at first by exactly definition of Hash1 and Hash2.

Common comments.

1.CE must be described.

2.Some of presented diagrams are trivial.

3.Presented key disclosure attacks are naive in the context of the paper.

4.The notion of bulletproof must be introduced and clearly defined.

5.The method of CE referenced in internal source [14] must be presented.

6.The confusion of programming language and math operations take place, e.g. formula (40) is incorrect.

Round 2

Reviewer 2 Report

The response to my comments is adequate and therefore I recommend to publish this paper.